# The Phase Modulating Micro-Mover Based on the MHD/MET System in the Reference Arm of the Scanning Interferometer

**DOI:** 10.3390/mi13111972

**Published:** 2022-11-14

**Authors:** Sergey Kalenkov, Pavel Skvortsov, Aleksandr Tarasenko, Dmitry Sharov, Alexander Shtanko

**Affiliations:** 1Scientific and Technical Center “Optoelectronics”, Moscow Polytechnic University, ul. Bolshaya Semyonovskaya 38, 107023 Moscow, Russia; 2Laboratory of Computer Systems for Production Automation and Digital Technologies, Mechanical Engineering Research Institute of the Russian Academy of Sciences, Malyj Haritonyevskij per. 4, 101000 Moscow, Russia; 3Department of Physics, Moscow State University of Technology “STANKIN”, 1 Vadkovsky Lane, 127055 Moscow, Russia

**Keywords:** digital hyperspectral holography, scanning interferometry, phase shifting interferometry, MHD actuators, MET sensors, electrochemical sensors

## Abstract

The possibility of using a magnetohydrodynamic drive (MHD) and amolecular-electronic transfer (MET) sensor as a single device for moving and precise control of the displacement of a movable mirror, which is part of a scanning interferometer, is considered. A prototype of such a device was developed and experimentally studied. A digital holographic image of the test object was obtained using an optical scheme containing a scanning interferometer with an MHD drive. The important advantages of the MHD drive in the problems of digital recording of hyperspectral holographic images have been discussed.

## 1. Introduction

Optical scanning interferometers—interferometers with a movable mirror in the reference arm—have been widely used in various branches of science and technology. One of the most important elements of such interferometers is the device for moving the mirror and precise control of its position. In particular, this concerns the problems of interferometry, digital holography, and microscopy. This is due to the fact that the change in the phase of the reflected optical signal is proportional to the displacement of the scanning mirror. As a result of of the large wave number (in the optical range), a small error in the position of the mirror leads to a large phase error, which, during further post-processing, seriously affects the quality of the reconstructed image. In particular, such interferometers are used in digital holographic microscopy—a branch of optics that was rapidly advancing in the early nineties of the last century. One of the first methods proposed for digital holograms registration was phase-shifting digital holography (PSDH) [1]. A number of interference patterns formed by the reference and the object waves are registered with a stepwise incrimination of the phase of the reference wave. Post-processing of these digital holograms allows the object wave field to be calculated. The key advantage of acquiring multiple phase-shifted holograms in comparison with a single digital hologram registration is the ability to suppress the conjugate image and zero diffraction order. A variety of PSDH methods have been proposed suggesting different amount of phase steps, including continuous phase shifting [2], when a large number of holograms are recorded [3]. Hyperspectral holography technique—the method of recording digital holograms in incoherent light [4,5,6]—is of particular interest. To clarify the requirements for a device for moving and controlling a mirror in the problems of recording and reconstructing digital and, in particular, hyperspectral holograms, consider the basic principles of hyperspectral digital holography below. The principle of operation of the proposed hyperspectral microscope is based on a new application of a Fourier transform spectrometer—a device originally designed for high-precision analysis of the spectral composition of electromagnetic radiation. The essence of the proposed method for registering hyperspectral holograms, described in [7], is that a flat micro-object is placed in one of the arms of an asymmetric Fourier spectrometer in place of a stationary mirror. The output signal, formed by the interference field of the reference wave with the wave diffracted on the object, is recorded by a multi-element sensor (CMOS). The registration process comprises the recording of a set of interferograms in each pixel of the sensor during a stepwise change in the length of the optical path in the reference arm of the interferometer. One-dimensional Fourier transform of the interferogram over the delta variable (optical path difference) in each pixel of the sensor gives the distribution of the complex amplitude of all spectral components of the hyperspectral object field at a given point. As a result of such processing of optical information, a two-dimensional spatial distribution of the complex amplitude of the object field for all spectral components, i.e., a hyperspectral hologram, is obtained in the entire array of pixels of the sensor. The positioning accuracy of the moving mirror largely determines the spectral resolution of the interferometer, the signal-to-noise ratio, as well as the quality of the reconstructed complex amplitude of the object field. A number of phase modulators are used for phase-shifting implementation: piezoelectric transducers, various magnetic actuators (voice coils), acousto-optic light modulators [8], liquid crystal light modulators [9], fiber-optic modulators [10]. These devices suffer from various imperfections and most of the time require precise calibration. Commonly, precise position control systems have to be integrated to increase the accuracy of mirror positioning. These circumstances urge to search for new phase-shifting devices for scanning interferometers. In this paper, we investigate the possibility of using magnetohydrodynamic (MHD) actuators and molecular-electronic transfer (MET) sensors and (MHD/MET system) to build up a micro-mover for hyperspectral holographic analysis applications. The advantages of the MHD/MET system are the absence of strict fundamental limitations characteristic of piezoelectric actuators, the fundamental uniformity of movement in contrast to the stepper motor and coil-actuator, as well as the ability not only to excite micro-displacements, but also to control them with high accuracy using the built-in MET sensor. Molecular-electronic transfer technology (MET) is based on the phenomena of mass and electric charge transfer in liquid electrochemical microsystems with characteristic sizes from units to several hundredths of a micrometer [11]. The main application of the technology is associated with the use in highly sensitive electrochemical seismometers and sensors for seismic exploration [12,13,14,15]. In recent years, many different technical solutions related to micromachine technologies have been created for this field of application and aimed at creating mass production of sensors [16,17,18,19,20]. There are also known results in the field of creating hydrophones, tilt meters, sensors for navigation, sports and medicine [21,22,23]. In addition, within the framework of this technology, it has been possible to develop not only highly sensitive sensors, but also microactuators based on the use of miniature electrochemical cells placed in a magnetic field and designed to create microflows liquids due to the magnetohydrodynamic effect of MHD [24]. In particular, such actuators are used in angular velocity and angular acceleration sensors [25,26].

## 2. Materials and Methods

### 2.1. MHD/MET System Design

To study the output characteristics of the mover of the mirror and the sensor, which provides a high-precision determination of its position, an experimental sample of a liquid drive for the mirror mover was created (Figure 1). The mover contained a housing made of inert plastic with a channel filled with a working fluid based on iodine–iodide electrolyte (KI concentration 4 M, iodine concentration 0.4 M). The channel contained a MHD cell including a pair of electrodes 1, placed in a magnetic field of permanent NdFeB magnets which generates liquid motion, MET cell 3, comprised of four platinum grid electrodes separated with dielectric partitions. The geometry of the electrodes and partitions is the same as that used in serial angular MET sensors as described in [27,28] designed to measure the liquid flow. Additionally, the channel was ended with flexible membranes. A mirror was attached to one of these membranes. The layout of the device is shown in Figure 1, right side. For this device, the amount of liquid flowing through the channel, due to its incompressibility, uniquely determines the movement of the membranes that limit those located at the ends of the channel. Moreover, even weak fluid motion could be detected and measured with high accuracy due to extremely high MET sensor sensitivity. In addition, in this design there is no dry friction and/or typical for stepper drivers nonuniformity of motion. An actual view of the MHD/MET system assembly is presented in Figure 2. To control the device, special electronic circuits have been made: a circuit for controlling the MHD device and a circuit that ensures the operation and reading of a signal from the MET cell. These circuits were implemented with galvanic isolation to avoid the flow of parasitic electric currents between the MHD cell and the MET cell electrodes. Figure 3 shows a current generator circuit to control an MHD device, where $I_{MHD}$ is the current supplied to the MHD electrodes and controlled by the voltage Ucur according to the following law: $I_{MHD}=U_{cur}/RX15$.

### 2.2. Study of the Frequency Response of the MET Sensor

To ensure the accurate determination of the position of the scanning interferometer mirror, it is necessary to calibrate the MET motion sensor, i.e., to establish a relationship between the linear displacement of the membrane mover and the output signal of the MET sensor. For this purpose, a stand was used, the diagram of which is shown in Figure 4. The main part of the stand—a balanced platform 1300 mm high, 600 mm long, 400 mm wide and weighing more than 60 kg—is suspended on a rigid torsion bar and is driven by speakers (2). The movement of the platform is controlled by position sensors (3). The rest of the parts include 4—electronic circuits of platform position sensors; 5—two-channel power amplifier; 6—electronic circuits of the automatic control system; 7—electronic board of digital-to-analog converter; 8—16-channel, 14-bit data acquisition system on the control computer bus; 9—control computer. The design eliminates parasitic oscillations of 0–320 Hz due to the selection of the rigidity of the elements. The calibration procedure can take place both in manual mode and automated, controlled by a digital-to-analog converter. The calibrated mover 10 is fixed steadily above the edge of the platform, the lower membrane is rigidly connected to the edge of the platform. Since the sensor readings depend on the throughput volume of the electrolyte, the stand is undemanding to the accuracy of the sensor mounting vertical, which simplifies the preparation for calibration. During the calibration, the stand performs sinusoidal oscillations (swing around the axis) with a frequency varying from 0.001 to 1 Hz, recording the position of the stand platform and the MET sensor output. The resulting characteristic is shown in Figure 5.

### 2.3. Study of the Frequency Response of the MHD Actuator/MET Sensor System

To determine the dependence of the MET cell signal current on the current passed through the MHD cell, the frequency response of this dependence was measured in the frequency range from 0.001 Hz to 1 Hz. For this purpose, through the MHD cell electrodes according to the scheme in Figure 3, a current was passed, which varied according to a harmonic law with an amplitude of 15 mA. The MET cell electrodes were connected to the circuit in Figure 6. The voltage Uout was measured at the output of the circuit. After that, the signal current at the output of the MET cell was calculated by the formula $I=U_{out}⁄R1$. The measurement result is shown in Figure 7. Figure 8 shows the frequency dependence of the ratio of the displacement of the membrane with a mirror fixed on it to the current flowing through the MHD cell $x⁄I_{MHD}$, obtained by dividing the ratio $I⁄I_{MHD}$ shown in Figure 7 by I⁄x from Figure 8. The resulting dependence allows to calculate the current required to move the mirror at a given distance.

However, when using aqueous solutions as the working fluid, there are limits to the maximum voltage that can be applied to the MHD cell electrodes. Experiments were performed, the purpose of which was to establish the permissible limits of the currents passed through the MHD cell, above which gas formation begins on the electrodes due to the hydrolysis phenomenon, and, accordingly, the maximum level of movement of the movable mirror. To increase the upper limit of the transmitted current, the volume of the MHD engine was filled with an electrolyte solution with an active component concentration of 0.4 mol/L, which exceeds the commonly used concentrations in the range of 0.01–0.1 mol/L. The experimental results are shown in Figure 8. The dependence of the voltage on the MHD cell on the electric current passed through the cell over time is shown, where the blue graph shows the voltage across the 10 Ohm resistor connected in series to the MHD cell and corresponds to the current passed through the MHD cell, and the red graph corresponds to the voltage on MHD cell electrodes. It can be seen that, in principle, it is possible to obtain currents up to 150 mA, although starting from a current near 100 mA, the voltage between the electrodes becomes unstable. Therefore, all subsequent experiments were 112 carried out with an inflow of less than 60 mA, which allows to provide stable operating conditions for the electronic circuit and mirror moving as could be calculated from the data in Figure 5c, this current corresponds to the ≈30 μm maximum displacement of the mover membrane.

At the same time, a study was conducted on the ability of the MET cell to measure the fluid flow velocity in the channel, and, consequently, the mirror movement velocity. The corresponding measurement is illustrated in Figure 5. In the MET cell (blue graph), when a current is passed through the MHD cell with a period of 10 s, the red graph shows the change in current flowing through the MHD cell. Similar to the measurements described above, the voltage measured across a 10 Ohm resistor connected in series with the MHD cell is presented. The presented graph shows, in particular, the absence of noticeable non-linear distortions, which qualitatively indicates the linearity of the entire path for converting electronics-current signals in the MHD cell to the output signal of the MET cell. At the same time, a study was conducted on the ability of the MET cell to measure the fluid flow velocity in the channel, and, consequently, the mirror movement velocity. The corresponding measurement is illustrated in Figure 9. In the MET cell (blue graph), when current is passed through the MHD cell with a period of 10 s, the red graph shows the change in current flowing through the MHD cell. Changes in the current through the MHD cell (blue curve, shows the voltage value measured across the 10 Ohm resistor connected in series with the MHD cell) and the voltage between the electrodes of the MHD cell (red curve). The measurements are described above, the voltage measured across a 10 Ohm resistor connected in series with the MHD cell is presented. The presented graph shows, in particular, Figure 5. The MET cell output signal (blue trace) and MHD cell current (red trace) show the absence of noticeable non-linear distortions, which qualitatively indicates the linearity of the entire path for converting electronics-current signals in the MHD cell to the output signal of the MET cell.

## 3. Experimental Verification

This section demonstrates the possibility of using the MHD/MET system as a micro-mover in hyperspectral hologram recording schemes and covers the approach to measure the maximum spectral resolution of the system achieved with the micro-mover in its current state of design. To address this problem, the stability of the displacement in time was estimated. The considerations given below are based on the results of our work (see, for example, [4,7]), which describe in detail the procedure of recording and restoring a holographic image of objects, based on recording of the intensity of the interference field of the object and reference waves. An optical setup comprising an interferometer was arranged with the micro-mover placed into the reference arm to introduce the phase shift. (Figure 9). The beam from the laser source (LS) is divided in two by the beam-splitter (BS) forming the reference and the sample arm. The mirror (M) mounted onto the micro-mover translator (T) is displaced at a constant rate of speed of 0.162 μ/s to introduce phase modulation in the reference arm. In the process of the continuous displacement of the mirror, a set of interferograms is registered in each pixel of the senor. The interferogram G(ξ,δ)—the dependence of the intensity in an arbitrary pixel ξ on the variable δ has two summands: the background—$G_{0}$ and the interference term $G_{int}\left(\xi ,\delta \right)$, which has the form:(1)$$\begin{array}{c} G_{int}\left(\xi ,\delta \right)=\int S(\sigma )[A(\sigma ,\xi )\;r\exp\left(-2\pi i\sigma \delta \right)\;+\\ A^{*}\left(\sigma ,\xi \right)r\exp\left(2\pi i\sigma \delta \right)]d\sigma , \end{array}$$
where rexp(2πiσδ) is the reference wave with the amplitude *r*, A(σ,ξ) is the complex amplitude of the subject field falling onto the sensor, and S(σ) is the spectral power density of the source. Substantial term is $G_{int}\left(\xi ,\delta \right)$ because the background $G_{0}=S\left(\sigma \right)\left({\left|A\right|}^{2}+r^{2}\right)$ does not depend on the variable δ and hence is further omitted. The integration is performed in the spectral interval Ω where S(σ) is nonzero. The Fourier transform of the $G_{int}\left(\xi ,\delta \right)$ interferogram over the variable δ yields a complex amplitude A(σ,ξ) at each spectral frequency σ, i.e., a hyperspectral hologram of the object field.
(2)$$A\left(\sigma_{0},\xi \right)=1/Lr\;\Delta \sigma S(\sigma_{0})\int_{-L/2}^{L/2}G_{int}\left(\xi ,\delta \right)\exp\left(2\pi i\sigma \delta \right)d\delta$$

The set of complex values obtained in each pixel ξ at the spectral component $\sigma_{0}$ according to Equation (Equation 2) form the complex amplitude of the object field $A(\sigma_{0},\xi )$ in the registration plane. The image a(σ,x) of the object was obtained by applying the inverse Fresnel transform $|IF\left[conj\left[A\left(\sigma_{0},\xi \right)\right]\right]{|}^{2}$ at a proper distance *z*.
(3)$$a\left(\sigma ,x\right)=|IF\left[conj\left[A\left(\sigma_{0},\xi \right)\right]\right]{|}^{2}$$

Under the stable conditions, the corresponding power spectrum ${\left|F\left[G(\xi ,\delta )\right]\right|}^{2}$ has a sharp maximum at a certain spectral component $\sigma_{0}$. For the present case of single-wavelength laser illumination A(σ,ξ) is obtained for a single frequency component and it facilitates the quantification of the spectral resolution of the system.

The fragment of eyepiece reticle 1.5 mm in size is an object (O) placed into the sample arm.

Each pixel of the camera sensor (C) captures the interferogram—the intensity of the interference pattern formed by the sample and the reference in the process of mirror displacement (Figure 10a). Frequency-time domain representation of the signal obtained by applying the S-Transfrom [29] shows that it takes about 20 s for the mover to reach standard conditions and move at a constant rate of speed during another 20 s. The square modulo of Fourier transform of the linear part of the interferogram acquired during this period of time is presented in Figure 10b. The length of linear travel was 3.7 μ. Despite this, the laser spectrum width is $10^{-6}$ nm, the theoretical limit of spectral resolution of the interferometer for the given path length of the reference mirror is 40 nm at λ=532 nm, the width of the spectral curve in Figure 10b at half height is close to this value. Figure 11 represents the object reconstruction performed in accordance with the Formula (3). Figure 11a gives amplitude image ${\left|a\right|}^{2}$ and Figure 11b phase image φ=atan[Imag(a)/Real(a)]. The size of the square is 0.5 mm, the line width is 0.01 mm. The lines are quite resolvable, which agrees with the expected diffraction resolution for a hologram aperture equal to 1/9. Illumination inhomogeneity across the object field is due to coherent noise, commonly intrinsic to coherent radiation.

## 4. Conclusions

A prototype of a translation device designed to displace the mirror in the reference arm of the interferometer is presented. The device is based on the phenomena of mass transfer and electric charge in liquid electrochemical microsystems. The electromechanical parameters of the system are investigated. The maximum quantity of the mirror displacement of 3.7 μm achieved in the voltage range of −3.3 V. The spectral resolution of 40 nm is archived. The amplitude and phase images of the test-object were reconstructed. The scope of the future work is to increase the rate of speed of the mirror displacement, as well as to equip the device with a feedback system.

## Figures and Tables

**Figure 1 micromachines-13-01972-f001:**
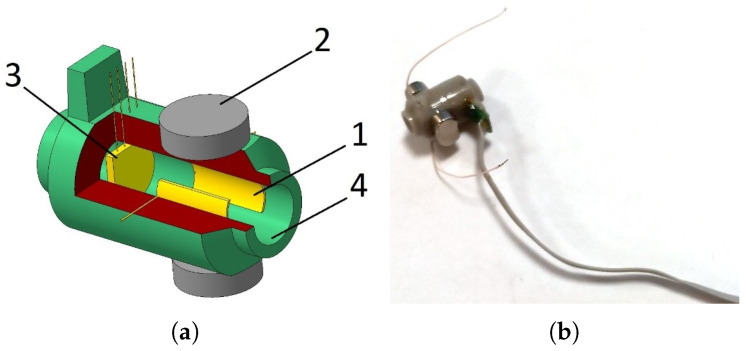
Schematic diagram and layout of the MHD/MET system. (**a**) The scheme, where 1 is pair of electrodes of an MHD cell, placed in a magnetic field of permanent NdFeB magnets 2, 3 is MET cell, 4 is glass ceramics channel; (**b**) The device.

**Figure 2 micromachines-13-01972-f002:**
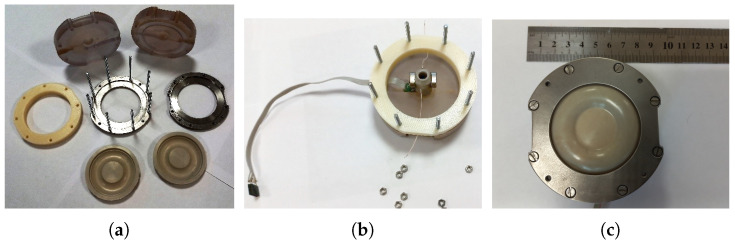
Photo of the MHD/MET system. (**a**,**b**) during and (**c**) after assembling.

**Figure 3 micromachines-13-01972-f003:**
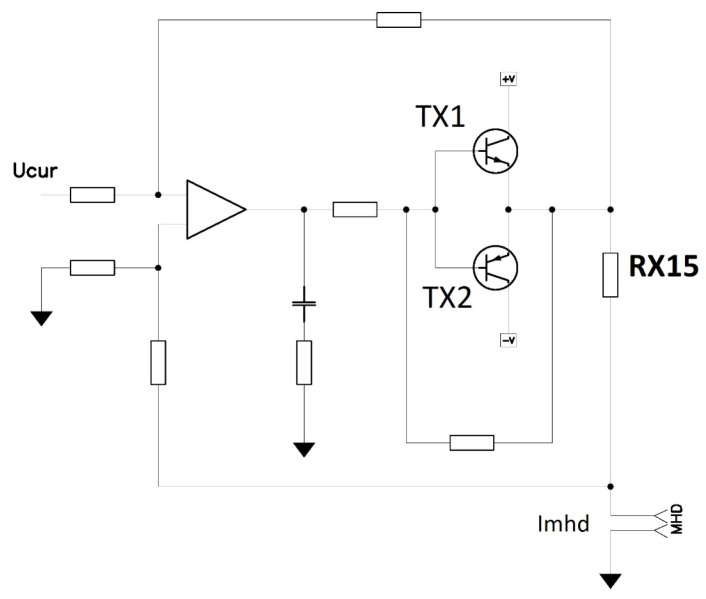
Scheme is electric current generator circuit with transistors TX1 and TX2 through the MHD cell. MHD are output pins for MHD electrodes.

**Figure 4 micromachines-13-01972-f004:**
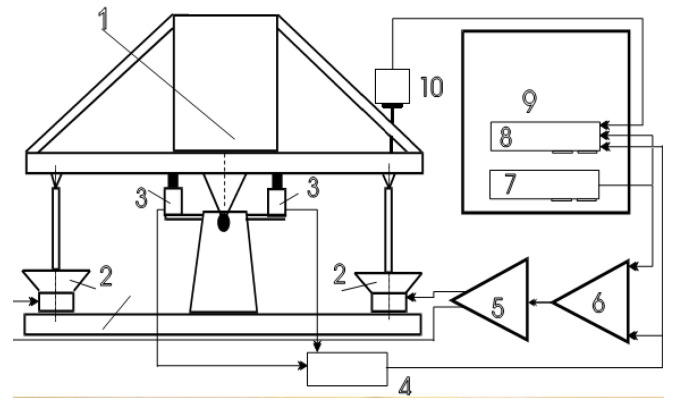
The principle structure of the MET calibrator of the displacement sensor.

**Figure 5 micromachines-13-01972-f005:**
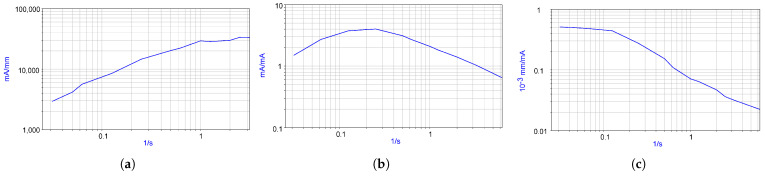
(**a**) Frequency response of MHD/MET system. The ratio of the signal current MET of the cell to the current in the MHD cell. (**b**) Ratio of cell signal current MET to mirror/platform bias. (**c**) Ratio of mirror/platform displacement to current in MHD.

**Figure 6 micromachines-13-01972-f006:**
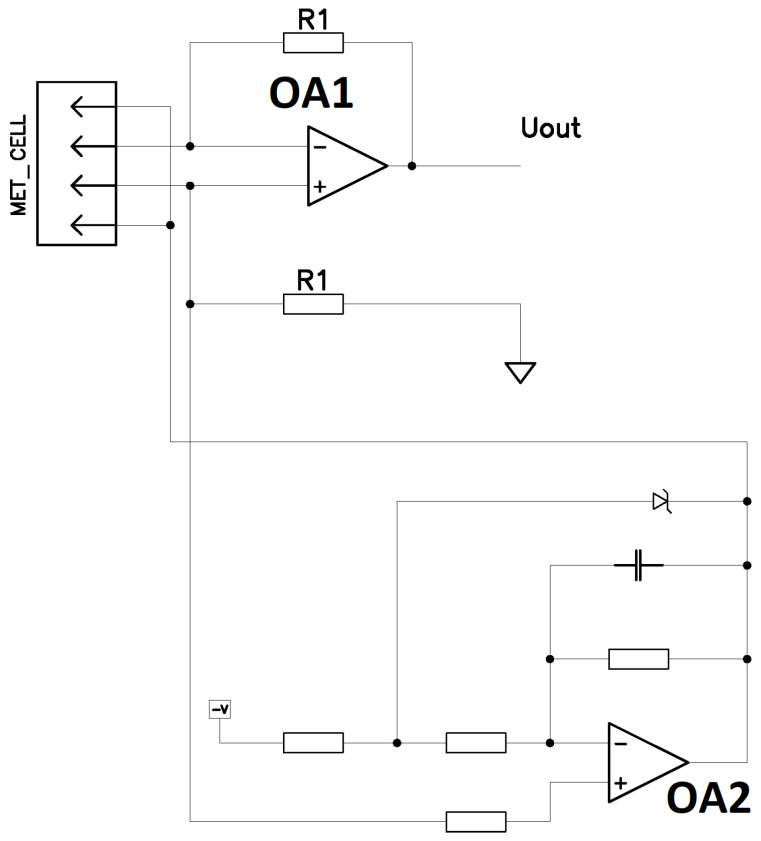
Scheme for electronics circuit of the MET cell. The MET cell in the figure denotes the pins connected to the electrodes of the MET cell. OA2 is operational amplifier providing potential difference between electrodes of MET cell. OA1 is operational amplifier transferring current from MET cell to voltage.

**Figure 7 micromachines-13-01972-f007:**
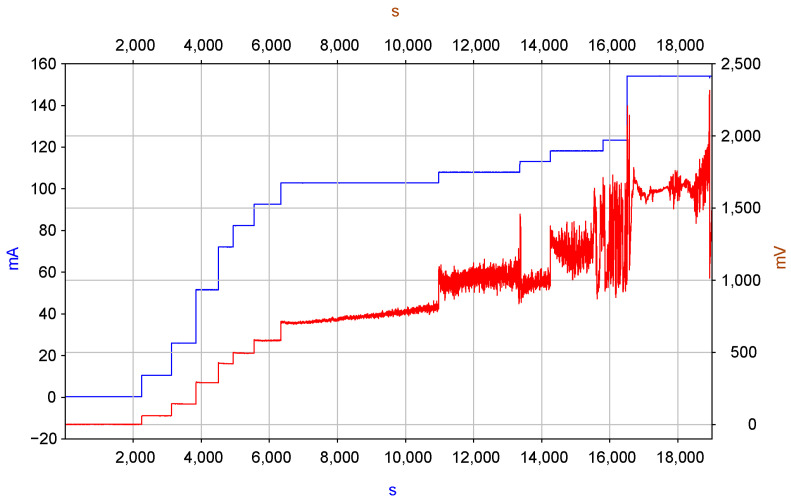
Changes in the current through the MHD cell (blue curve) and the voltage between the electrodes of the MHD cell (red curve).

**Figure 8 micromachines-13-01972-f008:**
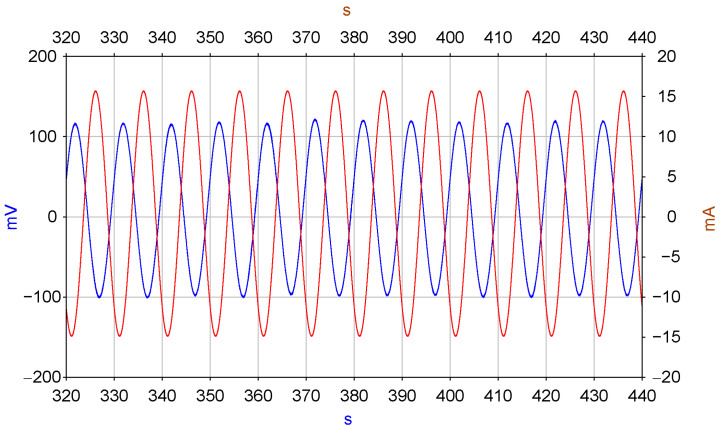
MET cell output signal (blue trace) and MHD cell current (red trace).

**Figure 9 micromachines-13-01972-f009:**
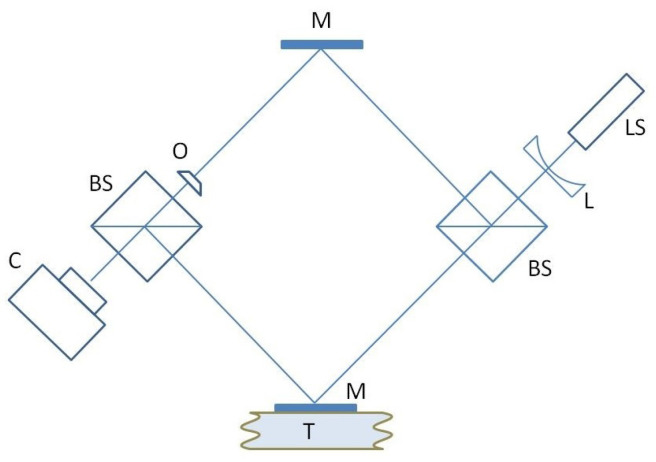
Optical scheme, registration of digital holograms; LS—laser source, L—lens, BS—beam splitter, M—mirror, T—translator, O—object, C—camera.

**Figure 10 micromachines-13-01972-f010:**
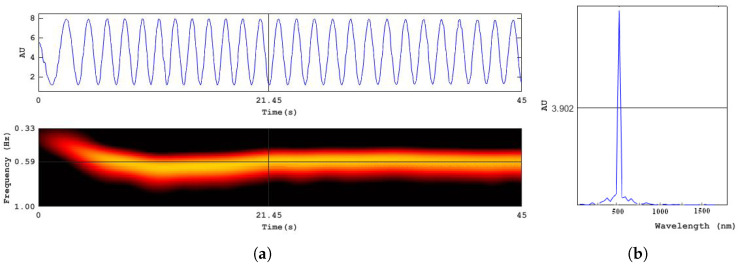
Motion of the reference mirror, spectral resolution of the interferometer; (**a**) An interferogram registered in a single pixel during mirror displacement; S-Transform of the interferogram, showing frequency shift due to nonlinear motion; (**b**) Spectrum of the linear part of the interferogram.

**Figure 11 micromachines-13-01972-f011:**
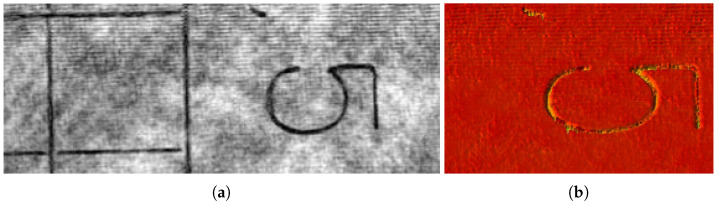
Object reconstruction (**a**) The amplitude image ${\left|a\right|}^{2}$; (**b**) Phase image φ=atan[Imag(a)/Real(a)].

## Data Availability

Not applicable.

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
