# Peer review of "The Phase Modulating Micro-Mover Based on the MHD/MET System in the Reference Arm of the Scanning Interferometer"

_micromachines, 2022, doi:10.3390/mi13111972_

Round 1
Reviewer 1 Report
1. Though the blue curve stands for the current passing through the cell, I suggest a secondary vertical axis in figure 5 to illustrate voltage and current on the same chart. At current state, the reader need to calculate the current value according to the voltage value and 10 ohm resistance. The same suggestion is for figure6.
2. In the sentence of "In the process of the continuous displacement of the mirror a set of interferograms is registered in each pixel of the senor", I guess "senor" should be "sensor".
3. Too much content is used to describe the scanning interferometer, more experiments should be implemented and more results should be illustrated to describe and verify the performance of the micro-mover.
Author Response
Authors thank reviewer for comments and suggestion to our manuscript.
Based on these comments we modified a manuscript.
Our point by point comments are added as separate Word document.

Reviewer 2 Report
See attached file

Author Response
We thank reviewer for very valuable comments and suggestions.
Based on that we significantly modified the document.
The detailed point by point response is added as a separate word document.
Besides , the English has been corrected by profissional interpreter.

Reviewer 3 Report
See uploaded file.

Author Response
Authors thank reviewer for very valuable comments.
Our point by point responses is presented in separate Word document.
Besides, the manuscript has been checked and corrected by professional interpreter.

Round 2
Reviewer 1 Report
no further comments
Reviewer 2 Report
The authors have taken my recommendations into account and provided an answer. I do not agree 100% but it would not be fair on my side to stick to some minor disagreements. Let this manuscript now be accepted for publication.